# Attracting, Recruiting, and Retaining Medical Workforce: A Case Study in a Remote Province of Indonesia

**DOI:** 10.3390/ijerph20021435

**Published:** 2023-01-12

**Authors:** Farah C. Noya, Sandra E. Carr, Sandra C. Thompson

**Affiliations:** 1Division of Health Professions Education, School of Allied Health, University of Western Australia, Perth, WA 6009, Australia; 2Medical Education Unit, Faculty of Medicine, Pattimura University, Ambon, Maluku 97233, Indonesia; 3Western Australian Centre for Rural Health, The University of Western Australia, P.O. Box 109, Geraldton, WA 6531, Australia

**Keywords:** medical workforce shortage, rural and remote, recruitment and retention, sustainable rural medical workforce, medical school initiatives

## Abstract

Indonesia, one of the Asia Pacific low-and middle-income countries (LMICs), has suffered from a chronic medical workforce shortage. However, there are limited published studies describing the approaches implemented by the Indonesian government regarding the recruitment and retention of the medical workforce. This case study aimed to understand the current practices for recruitment and retention of the medical workforce in Indonesian rural and remote provinces. We conducted a case study of the Maluku Province of Indonesia with a document analysis and key informant interviews with officials responsible for medical workforce recruitment and retention. We used the World Health Organization’s (WHO) guidelines as an analytical matrix to examine the recruitment and retention practices under the four domains of (i) educational, (ii) regulatory, (iii) financial, and (iv) professional and personal development and classified them into either University/Medical School level or Government/Non-government level. Our findings suggest that Indonesia implemented most of the WHO-recommended medical workforce recruitment and retention strategies. However, implementation is still problematic; hence, the aim of establishing an adequate, sustainable medical workforce has not been reached. Nationwide government intervention in educational aspects is important to magnify the impact of regional medical school initiatives. Relevant programmes must be re-evaluated and re-enforced concerning significance, comprehensiveness, and effectiveness for a sustainable rural and remote medical workforce.

## 1. Introduction

Medical workforce shortage in rural and remote (RR) areas has been a significant problem globally. Since the introduction of the World Health Organisation (WHO) Global Policy Recommendation towards improved retention [1], more nations globally have implemented recommended strategies. Of those strategies implemented, educational strategies involving the active contribution of medical schools were documented as effective for improving recruitment and retention of the medical workforce [2]. Admission of students with a rural background, rural immersion, and comprehensive medical school programmes were the most effective strategies reported over time and across places [2]. These strategies documented early and continuous interventions, with the term Rural Pipeline used to describe this combination of strategies [2]. Some low- and middle-income countries (LMICs) in the Asia Pacific, such as Thailand, have realised improved recruitment and retention of rural medical workforce as a result of the implementation of educational strategies [2].

Indonesia, one LMIC in the Asia Pacific, has suffered from a chronic medical workforce shortage [3,4]. Although Indonesia barely achieves the national standards for the underlying health indicators [5], there are limited published studies describing the approaches implemented by the Indonesian government regarding recruitment and retention of the medical and healthcare workforce. Health workforce strategies operate concurrently with other public health strategies nationally or regionally, which aim to improve population health. This study focuses on the Maluku province of Indonesia, which includes some of the most remote, isolated, and underserved islands [3,5]. The remote and isolated nature of the province is reported as having discouraged physicians from living and working there [6,7,8]. With less health infrastructure and fewer facilities, the ratio of physicians per population is 1:7269.

The districts in Maluku are districts where health service facilities are determined as remote, very remote, and unattractive [9]. They are underdeveloped (73% of the province districts in Maluku fall under this classification) [10], located in the outermost islands (17% of national numbers) [11], and include challenging conditions being small islands, island clusters, or coastal regions characterised by poor public transport and difficult access with long travel times to the district or city capital, as well as where travel may be hindered by the climate or weather. Residents may experience difficulties in fulfilling essential commodities and have unstable security conditions [9,12].

The purpose of this study is to document and understand strategies recommended and implemented for building a sustainable medical workforce, including the constraints on the implementation of rural recruitment, development, and retention strategies in one remote province of Indonesia. This case study of a RR province in the eastern Indonesian archipelago, Maluku, with a qualitative review of government documents and interviews with key informants, allows for the triangulation of data to better understand the strategies currently implemented and challenges encountered in recruiting and retaining the RR medical workforce.

## 2. Materials and Methods

A case study is a qualitative study method that intensely explores an issue through one or more cases within a bounded system through comprehensive data collection to understand a complex issue in its real-life context [13]. Document analysis is often used simultaneously with other qualitative research methods for data triangulation of a particular phenomenon [14]. It is expected that data are drawn from more than one source and inform each other. In this case study, we drew comprehensive data from a document analysis [14] and interviews [15,16].

### 2.1. Document Analysis

We collected documents published online or for internal use from the national, provincial, and district government officials regarding strategies for recruitment, development, and retention of the medical workforce. The documents were national policies relevant to the recruitment and retention of the medical workforce, provincial and district health strategic plans and human resource planning, medical school strategic plans, accreditation documents, and distribution data of students and graduates. Other data sources were international, national, and regional published news and articles regarding the recruitment, development, and retention of the medical workforce in Indonesia and Maluku.

The researcher accessed Maluku provincial government documents through a government website and with permission from government officials. Documents from the regional medical school, Pattimura University Faculty of Medicine (PUFM), were accessed through school executives. Data collected from the documents related to the strategies implemented for the recruitment, development, and retention of physicians in RR areas.

### 2.2. Key Informant Interviews

The key informants (KIs) were identified and listed based on their background, experience, present involvement with the medical workforce and related areas, and potential to provide valuable information on rural retention of the medical workforce across the province.

The KI interviews (KIIs) collected data regarding recruitment, development, and retention strategies in the province and districts. Using an interview guide, we conducted interviews with 15 respondents, including provincial health officials related to human resources (*n* = 3), heads of District Health Offices (*n* = 7), the dean and former dean of PUFM (*n* = 2), the head of the Regional Indonesian Medical Association (*n* = 1), and *Puskesmas* (Community/public health centre at the sub-district level) accreditation surveyors (*n* = 2). The questions posted related to the implementation of strategies for recruitment, development, and retention of the medical workforce, including the challenges experienced.

### 2.3. Data Analysis

A thematic analysis [17] was used for data synthesised from the document analysis and the KIIs. The themes were predefined in the analysis framework. The analysis framework was based on a combination of the framework from our scoping review of strategies for improved rural and remote medical workforce [2] and the WHO Global Policy Recommendations [1]. The previous added levels of implementation to the latter, i.e., University/Medical School level and Government/Non-government level.

We extracted the strategies documented in the government and medical school documents and derived from the interviews and then classified them into two levels according to the framework: (1) University/Medical School and (2) Government/Non-government (National—except for the data from key informant interviews, Provincial, District) [2]. Then, we mapped the strategies under each theme (and subtheme) from the WHO recommendations: education, regulatory, financial incentives, and professional and personal support [1]. The numbering used for themes and subthemes adhered to the WHO recommendations numbering [1] (i.e., A1–D6. Table A1). Missing number means that the particular strategy was not implemented/applicable at that level. In this study, we reported the interventions based on the predominant aspects implemented by the stakeholders. Challenges faced in each strategy were reported in the respective themes and subthemes. We used narrative reporting of the interviews to convey the intellectual and interpretative message from respondents [18], including some verbatim quotes with alphanumeric codes (KI#, means Key Informants’ number) to express the challenges faced.

### 2.4. Trustworthiness

The informants were assured of the anonymity and confidentiality of their information and were allowed to talk without being recorded on issues they considered sensitive. Verbatim transcription in the Indonesian language was performed on the audio-recorded interviews and later translated to English to enable cross-analysis and verification by other researchers. The researchers had experience in medical education and the rural medical workforce that could influence the interpretation of data, but throughout maintained reflexivity and followed the analysis framework. The data extraction, coding, and interpretation adhered to the predefined themes of the analysis framework.

### 2.5. Ethics

Ethical approvals to undertake this study were granted by the University of Western Australia Human Research Ethics Committee and the Pattimura University Research Ethics Committee.

## 3. Results

### 3.1. University/Medical School Level

At the national level, apart from the Ministry of Health (MoH) initiative to accelerate health profession education to reach minimal qualifications [19], there was no educational intervention found or imposed by the Ministry of Education for universities nationwide for medical workforce recruitment and retention. Some medical schools (42 of 83) located outside Java had programmes related to educational recommendations, but this was not related to a systematic agenda imposed by the Ministry of Education [20]. PUFM, the only medical school in Maluku Province, was established in 2008 with a mission to fulfil the province’s needs for a medical workforce and better quality of health services [21,22]. At the PUFM level, educational interventions were employed. Based on both the documents and interviews, the strategies implemented in PUFM complied with all WHO educational recommendations (A.1.–A.5.) and regulations regarding compulsory service (B.3.) by requiring its alumni to undertake a minimum of one year of compulsory service in RR districts.

#### A. Education


*A.1. Get the “right student”*


The strategy of student selection for rural background and interest was implicitly documented in faculty strategic planning and accreditation documents [21,22]. There were memorandums of understanding with three districts in the province to select students from the districts for medical education at the PUFM [21,22]. Although PUFM had no stated policy or quota for rural background students, each year it admitted 20–25% of students from rural and remote districts, both districts without or with the memorandum of understanding (MoU) with the faculty [3,23]. The selection was conducted on two levels, first by the district’s officers for screening purposes and later by the university. However, the selection procedure was not documented in detail.


*A.2. Train students closer to rural communities*


PUFM is located in the remote province of Maluku, far away from the capital of Indonesia. Indeed, Maluku is the second-farthest province. This means that the students were trained close to the communities they would serve upon graduating. The faculty had partner communities and *Puskesmas* where the students partook in their training during their six years (four years of preclinical stage and two years of clinical rotation) of undergraduate medical education [21,22] (related to recommendation A.3.).


*A.3. Bring students to rural communities*


Related to recommendation A.2., the PUFM curriculum arranged community placements for medical students in the preclinical and clinical stages. Some were in *Puskesmas*, communities and floating hospitals across the islands, where the students undertook clinical rotations (public health and community medicine for 6 weeks), preclinical community service (in year 4 for eight weeks), and faculty community service programmes [21,22,24]. The faculty also had several communities where research, service, and education were devoted to developing those communities in health. The students were immersed in these communities, living and learning together. However, the number of these communities was limited, and their location was only in rural areas near the province’s capital [21,24].


*A.4. Match curricula with rural health needs*


The PUFM curriculum documents reflected a rural focus, especially within the archipelagic context of Maluku province [21,22,25]. Similar to other Indonesian medical schools, the PUFM undergraduate curriculum was delivered in blocks. During the preclinical stage, there were compulsory blocks that focused on rural health, and during the clinical stage, a 6-week clinical rotation focused on rural health [25]. The rural-focused learning objectives were predefined in units, blocks, and learning topics with local contents to bring the learning to the rural community context, especially the archipelagic context of Maluku [21,22,25]. Faculty research and service programmes were also mapped under the vision statement to provide community health services and address health problems in the Maluku islands [21,22].


*A.5. Facilitate professional development*


The PUFM documented its strategy regarding facilitation for professional development through programmes such as seminars, specialist consultations, and workshops dedicated to alums and the broader medical workforce in Maluku province, including telemedicine [21]. Together with the Regional Indonesian Medical Association, the programmes assisted and facilitated Maluku RR physicians’ professional development.

#### B. Regulatory Intervention


*B.3. Make the most of compulsory service*


Pattimura University Medical Faculty required its alumni to undertake a minimum of one year of compulsory service. This aimed to distribute the medical graduates to more rural and remote areas in Maluku Province, especially to the districts that send their students to be trained at PUFM. Although this intervention was enforced on the graduates, it was not documented in strategic planning or other faculty documents.

KIIs revealed that the alumni were required to sign a written statement at the inauguration indicating that, after the internship, they would participate in temporary employment programmes in any district in Maluku province.

We oblige our alums to take temporary employment for one year in an effort to deploy personnel in the archipelago. Thus, they must sign a written statement at the inauguration indicating that after the internship, they are willing to take part in temporary employment in any district in Maluku province.(KI#9)

Those who received regional scholarships and recommendations must return to their respective districts, while those without a regional recommendation or scholarship must go to the districts that required physicians. KIIs with district health offices confirmed this arrangement and the resultant benefits. The PUFM supported its graduates for compulsory services by facilitating workplaces and continuing medical education programmes.

### 3.2. Government/Non-Government Level

Strategies found at the national level related to regulatory intervention, financial incentives, and personal and professional support. We found data regarding the numbers of physicians at the district, provincial, and national levels. At the district level, limited documents provided data about strategies implemented or planned to improve rural and remote physicians’ recruitment, development, and retention. The strategic planning of the province and district health offices acknowledged the low distribution and shortage of medical workforce as one of the strategic issues affecting the population’s health status. Despite this, there was no mention of programmes to execute or those being implemented to improve the quantity and quality of human resources, including physicians in RR areas. Most strategies were revealed from interview data.

#### B. Regulatory Intervention


*B.1. Create conditions for rural health workers to achieve more*


Regarding the enhanced scope of practice, the MoH and the Indonesian Council of Medicine in 2011 agreed to train general physicians with additional competencies outside their prescribed ones, known as ‘General Physician Plus’ [26]. This was to respond to the challenges of achieving the Millennium Development Goals to significantly reduce maternal and child mortality in areas of Indonesia with maternal mortality rates and infant mortality rates above the expected level, including Maluku [26,27]. The competencies included but were not limited to ‘performing caesarean section’ and ‘performing abdominal ultrasonography’ [26]. However, the MoH did not issued a regulation to legalise the agreement, and the training by relevant collegium was ceased due to a tight budget [26].

At the provincial level, Maluku had a programme called ‘*Gugus Pulau*’ or ‘Islands Clusters’, planned to solve geographically remote islands’ health problems [28]. This island groups’ health service was created to bridge the geographical, economic, and socio-cultural disparity within Maluku Province. This programme focused on building activities with the principle of independence so that one cluster centre could solve health problems within a cluster without unnecessary referrals. The KIIs revealed that there was still a lack of human resources for healthcare, especially physicians in the island-based cluster centres. However, there was little acknowledgement of medical workforce shortage as one main challenge why health services were struggling in Maluku Province.


*B.2. Make the most of compulsory service*


At the national level, efforts to distribute the physicians were carried out through mandatory work as a service to the country to improve the quality of health services so that the community had access to medical services, especially in rural, remote, disadvantaged areas, borders, and islands. Three programmes initially utilised compulsory service of physicians with allocated funding: ‘temporary employment’, ‘utilisation of medical specialist graduates’ (initially compulsory, both programmes changed over time and became voluntary [29,30,31,32]), and ‘medical internship.’

Temporary employment (Since 1961, *Pegawai Tidak Tetap* and *Nusantara Sehat*, 2018). The Indonesian National Health Policy regarding temporary employment emphasised the continuing availability of service in all divisions of healthcare workers in remote and underserved areas [29,32,33,34,35]. In addition to an incentive package, the *Nusantara Sehat* participants were given the opportunity to become civil servants and could apply for scholarships for postgraduate training after completing the programme. However, the retention rate of this programme has not been reported and requires further evaluation.Utilisation of medical specialist graduates. The utilization of medical specialists who made the choice to work there was substituted for the previous mandatory work of medical specialty graduates [30,31]. In practice, the mandatory work implemented since 2016 was seen as violating the physicians’ human rights and the International Labour Organisation policies regarding forced labour. The mandatory work was then replaced in 2019 by more conventional work arrangements. The MoH opened recruitment to specialist graduates, offering incentives and other benefits. However, the non-mandatory nature of the employment led to a decrease in the number of specialist physicians serving in rural and remote areas, especially in the eastern regions, including Maluku [36].Medical internship. Although it is not primarily aimed at rural and remote medical workforce distribution, medical internships partly contributed to the dissemination of the medical workforce across RR districts in Indonesia [37,38].

At the district level, the districts that provided medical students with financial support imposed official demands. The students were obliged to return to and work in their respective districts. The PUFM supported this implementation with signed written statements by the newly graduated alums to return to their districts. The districts supported these physicians with financial incentives and recommendations for postgraduate training (specialists) in addition to permanent positions as civil servants in the districts. Furthermore, the districts and provincial government also recruited physicians with local employment schemes, both temporary and permanent, with differing numbers across districts. However, that physicians were attracted to RR places in the districts through temporary employment schemes had a downside where those RR areas only had the physicians for a short time [3].

The KIs reported that, while the MoH compulsory and voluntary schemes were employed simultaneously with districts’ efforts to recruit medical graduates of local origin and with local funding schemes, many *Puskesmas*, especially in the remote islands, were still vacant. For example, in one of the districts, only 63% of the Puskesmas were occupied by physicians.


*B.3. Tie education subsidies to mandatory placements*


At the national level, the MoH had policies regarding mandatory placements for physicians who the government assisted through educational funding aid for their specialist training. They were required to serve in hospitals in remote areas that needed specialists with a time bond minimum of n years and a maximum of 2n years (n = length of study, which varied depending on the specialist training program) [20,31,39,40]. This regulation increased the number of specialist physicians in rural districts [20]. However, it could not increase the number of physicians in remote, very remote, and bordering areas, as specialist physicians could only work in the hospitals in district capitals. Another downside of this programme was that only 73% of the specialist physicians with scholarships or financial support returned to their respective districts [20]. As mentioned by a KI, in their district some physicians failed to return without notice, and their locations were not always tracked.

We have physicians trained with scholarships from the MOH through the district’s recommendation. They haven’t come since they graduated. One has practised in Bekasi (West Java). The MOH should fine them because MOH funded their training. But, the MOH did not give any sanctions. At least hold their registration, so they can’t practice elsewhere. Well, the supervision did not work, or maybe there was someone inside, I mean, like collusion, helping them with the registration, which made the person, didn’t return (here).(KI#1)

Aru Islands, Buru, and South Buru are three districts in Maluku with memorandums of understanding with the PUFM to send their local people to medical school [21]. Annually, these districts selected 5–15 students and sent them to the PUFM selection process. Those who succeeded in the selection were fully funded until they completed a medical degree. Some other districts did not hold a selection process but did fund students from their area who had already been admitted to the PUFM. These students were partially or totally funded for tuition fees and living expenses. In these districts, the compulsory return of services was made possible with the educational subsidies provided by the government. The medical graduates returned to their respective districts as per the agreement they made with the districts. PUFM-enforced regulation of mandatory service also directed the graduates to their respective districts. Some districts set time limits for the return services. Other districts sought to retain physicians who had already become civil servants for as long as possible, even throughout a physician’s career, by providing scholarships or financial assistance.

When we have determined to accept (civil servants), we have warned them they cannot move and must stay here. We have a lifetime contract. Specialist physicians here were general physicians at the Puskesmas. We send them to be trained as surgeons, internists, obstetricians (etc.). Now (they have) come back. Most are civil servants. We pay the salaries of civil servants, and we provide an incentive.(KI#3)

#### C. Financial Incentives: Make It Worthwhile to Move to a Remote or Rural Area

The Indonesian government, through the MoH, provided financial incentives to physicians serving RR Indonesia under national voluntary or utilisation schemes. Salary and incentives packages to temporary physicians were higher than those with permanent positions or employed within other schemes. To increase the interest of physicians in participating in voluntary or utilisation schemes, temporary physician employees received almost twice as much as permanent or government physician employees [29,41]. However, for the medical internship, the salary and incentives known as Basic Living Expenses were smaller and deemed to not meet the necessities of life compared to the volume of work assigned to the interns [41].

At the district level, in addition to the support given to their physicians with bonded placements and local employment schemes, most districts also provided incentives for physicians from national employment schemes, such as Nusantara Sehat and temporary specialists. Hence, these physicians with national appointments received extra incentives. There were also discrepancies in the incentives provided for physicians across different districts. As exposed from the KIIs, the provision was mainly dependent on the locally generated revenues. The nominal range was between Indonesian Rupiah (IDR) 2.5–10 million for general physicians and IDR 25–35 million for specialists per month. Districts with higher incomes were able to place more budget to attract physicians, while those that were poorer mainly depended on the MoH regulation for temporary employment. Those who gave more to the physicians demanded their service, whatever it took. The districts expected that they already provided significant incentives and service funds, so physicians were obliged to serve, whatever the living and working conditions.

They [the physicians] get incentives and service fees. They get houses. All specialist physicians get cars. I don’t think there is a reason anymore not to stay. For physicians [general practitioners] in remote islands, they can get incentives, they get the 60% of JKN (National Health Insurance) service fees from there, and they can still earn more if (patient) is hospitalised (transfer to the higher-level facility). They got Health Operational Assistance funds and transportation. Yes, it is pretty enough.(KI#3)

Some of the KIs mentioned the adverse effects of this strategy. Although districts already provided significant incentives, some physicians did not stay for the duration of service. They did not like being in remote and isolated areas, especially those originating outside Maluku. The amount of money spent to retain the physicians, therefore, failed to achieve its purpose. However, some districts commended the Nusantara Sehat and locally contracted physicians’ work.

Another financial incentive came from national health insurance funds in the form of service fees. This incentive was provided to all serving health practitioners, including medical interns and physicians from diverse employment schemes [42]. Furthermore, there was a policy in which the insurance provider increased the amount of capitation for *Puskesmas* with better performance [43].

In terms of transport and housing, not all physicians received the same provision. As exposed by the KIs, some districts provided vehicles and built their physicians’ houses to retain them for longer. However, this was more difficult for some districts, especially in remote areas.

#### D. Personal and Professional Support


*D.1. Pay attention to living conditions*


Nationally, there were programmes for improving living conditions in the RR areas of Indonesia. Stakeholders of these programmes were the Ministry of Finance, the Ministry of National Development Planning, and technical ministries such as the Ministry of Public Works, the Ministry of Health, and the Ministry of Villages, Development of Disadvantaged Regions, and Transmigration. One of the focuses of these programmes was the acceleration of development in eastern Indonesia, including Maluku [44]. ‘Dana Desa’ or Village Fund was one of the featured programmes with funding prioritised for villages’ basic needs (health and early education) and the development of village facilities and infrastructure [45]. There were also the ‘Balancing Funds’ sourced from national revenues such as ‘Special Allocation Funds’ [46], which were allocated to certain regions intending to help fund specific regional activities in accordance with national priorities (education, health, roads, irrigation, drinking water, government infrastructure, marine and fisheries, agriculture, and the environment).

At the district level, there was documentation of the link down from the national government programme ‘Dana Desa’ (Village Fund) [47,48,49,50,51,52] and other national development plans [46]. However, as exposed in the KIIs, many districts and villages did not benefit from the programmes. Despite the complete cycle of the use, monitoring, and evaluation of the programmes mandated by the national rules [53,54], necessities such as clean water, electricity, road access, and transport in RR areas were still scarce.

Well, another thing that makes the physician more comfortable is, for example, clean water resources that support their lives. There are no good water resources available. Most villages have no electricity, but at the Puskesmas, we facilitate with a generator. But, then the problem is with the fuel, no fuel in the villages. Going to the city will take more money (far distance, sea transport with rare fuel).(KI#2)

Government institutions did not seem to work in unison for improvements. Another example, the district health office provided a vaccine freezer for Puskesmas, but there was no electricity. In other districts, health offices provided power generators for remote Puskesmas with no electricity or ambulance boats for patients’ transport, but fuel availability was another problem. Moreover, as massively reported in the local news, there were corrupt practices by the villages’ officials responsible for the Village Funds [55,56,57].


*D.2. Ensure the workplace is up to an acceptable standard*


There was a fund provided through the MoH for the procurement and upgrading of health facilities and infrastructure through ‘Special Allocation Funds’ and ‘General Allocation Funds’ from the national revenues [46]. As confirmed by the KIs, the health offices in districts were entitled to request acceptable standards of workplaces with this allocation. This considerable amount of funds covered the construction of the Puskesmas and official housing and procuring health equipment, facilities, and furniture. These funds needed to be applied for through the Medical Device Infrastructure Application [58], which demanded descriptions of baseline data and the upgrade requested. According to the KIs, not all district officials understood and were able to prepare the proposal, needing more time for the application. Another reason stated was due to the use of the information system and reporting via a web-based application, so there was no chance of scamming and modifying a project value. Hence, while decreasing the chance of corrupt applicants, it likely also reduced applications, despite the critical needs. Many health facilities in RR Maluku remained with substandard working conditions.

They must fill out an application called ASPAK (Medical Device Infrastructure Application) to get the funds. It describes the actual condition of our health facilities’ medical equipment and infrastructure, so there is no fake data. For example, Puskesmas requires major renovations. Then, this need must be supported with photos of the particular Puskesmas. It is not the health sector’s responsibility to state whether the Puskesmas needs severe or light renovation. The Public Work Service shall determine the level of renovation required based on records and analysis about the Puskesmas. That’s the way it goes.(KI#8)

In their strategic planning documents, some districts reported programmes for improving facilities and infrastructure related to work, such as the construction of physicians’ accommodations, the procurement of operational service vehicles, and the procurement, maintenance, and upgrading of work facilities sourced from district revenues. From the KIIs, some districts acknowledged that they had not given optimal support regarding transport, housing, and furniture for the physicians working in their districts. Some mentioned the complicated bureaucracy regarding procurement. However, for other districts, medical service was seen as an obligation or a calling, so physicians were expected to serve whatever the working condition in the RR areas.


*D.5. Facilitate knowledge exchange*


The provincial health office provided personal and professional support through programmes such as the provision of clinical internship sites and training centres for physicians [28]. Although a recognised strategic issue was low quality and quantity of health human resources (HRHs), including physicians, there was little HRH planning and training data [28]. As informed in the KIIs, training and development programmes were usually tied to a region’s health improvement programmes, often linked to nationally prioritised programmes, and were not generally provided to all employees or physicians, especially in the remote islands.

At the provincial level, there was a Regional Indonesian Medical Association to facilitate knowledge and skills exchange for physicians working across the province. The KIs informed that the medical association often provided training, seminars, and workshops for physicians from all over Maluku but also dispatched physicians and specialists from the province’s capital city, including physicians from PUFM, to train and teach physicians in RR areas. Some districts had an active medical association; some had inactive or no support.


*D.6. Raise the profile of rural health workers.*


The Indonesian government, through the MoH, provided awards for health professions, including physicians, especially those who worked in Puskesmas for primary care, with recognition of them as ‘Exemplary Health Personnel’ [59]. This award was implemented at the provincial and district levels in Indonesia as a law mandate. In their activity plan, some districts documented ‘Exemplary Health Professionals’ selection as a programme for the capacity building of personnel.

Overall, there were documented achievements at the University/Medical School and Government/Non-government levels, but there were also areas needing improvement, as summarised in Table 1.

## 4. Discussion

This study found limited documentation at the district and provincial levels regarding strategies to improve recruitment and retention of the medical workforce despite the acknowledgement of shortages. Our findings indicate that Indonesia, particularly Maluku province, implemented most of the WHO-recommended strategies. However, the implementation approach taken did not achieved the desired outcomes. Many programmes were still not comprehensive, and the educational approaches applied were limited to medical schools.

### 4.1. Education

We did not find any education strategy, program, or regulation that was implemented nationwide. However, PUFM directed their education, research, and service activities towards addressing the priority health concerns of the community in Maluku Province [60]. It is likely that, as a result of interventions implemented by the PUFM, there were increased numbers of physicians working in RR Maluku [3]. The PUFM work was exclusively motivated by their internal philosophy and commitment as a provider of the medical workforce in Maluku Province. A more systematic solution to the chronic national shortage of the medical workforce is needed through the nationwide implementation of educational interventions.

Another LMIC country, Thailand, has implemented an educational strategy as part of their national regulation. Two government-funded projects have been employed through a collaboration between the Ministry of Education and the Ministry of Public Health [61]. With regulations on job placement, the duration of mandatory service, and non-adherence to obligation penalties applied, Thailand has increased their number of RR physicians significantly [61]. Learning from Thailand’s success and that of developed countries such as Australia and Canada in educational interventions, Indonesia should consider applying a nationwide educational intervention. Rural Clinical Schools across Australia were implemented as a national educational intervention to address the geographical inequalities of the medical workforce [2,62]. At the time, in terms of rural training locations, exposure, and rurally-focused curriculum, only a limited number of medical courses in Australia were located entirely in regional areas [63]. The advent of the Rural Clinical School programmes meant that medical schools were not acting sporadically and exclusively, but rather worked systematically under the considerations imposed by the national government. The predefined quota of rural background medical students in the admission programme ensured the representativeness of rural communities and supported the ‘rural pipeline’ programme [2,64]. In the end, this approach could benefit a community by having medical graduates that were more likely to return and be retained in the community.

### 4.2. Regulation

Our study found that temporary employment with its benefit from the MoH partially improved the recruitment of physicians to RR areas. There were also provincial and district efforts in recruiting physicians through local temporary employment schemes to RR areas. However, there was no documentation of an increased retention rate. Currently, the ratio of physicians in Indonesia is only 3.8 physicians per 10,000 population [65], and Maluku only has 1.5 general physicians per 10,000 population [20]. Hence, this suggests that physicians may be deployed to RR districts, but retention has its challenges. This case study confirmed a reluctance of the physicians to return, while the penalty obligation was not enforced. Many countries, such as Thailand, have successfully improved their RR medical workforce through imposed penalties for non-adherence [61]. This highlights the need to reinforce the regulation while addressing the main cause of the reluctance.

Other aspects of regulatory intervention were that no existing government regulation supported enhancing the scope of practices. The ‘General Physicians Plus’ programme [26] had the potential to be reconsidered to offer support through thoughtful regulation. At the moment, the strict regulations of the Indonesian National Health Insurance (BPJS Kesehatan) regarding physicians who are entitled to receive service incentives [42] may limit the efforts for such task-shifting-related strategies. This aspect highlights the need for deliberative efforts by the government in implementing this strategy.

### 4.3. Financial Incentives

Although offering financial incentives may attract physicians to RR places, they were found ineffective in improving the retention of physicians. A study in Maluku in 2021 found that salaries of more than IDR 6 million were associated with an 11-fold increase in rates of RR practice [3], confirming the importance of Indonesian government support through financial incentives included in the salaries of those who practice rurally within temporary assignment schemes [8,29,66]. However, given the temporary status of many physicians, a higher incentive appeared unlikely to guarantee the retention of physicians in RR practice [3]. Moreover, the difference in the amount of the financial incentives across districts and employment schemes despite the same workload created an unfair situation. This was likely to influence physicians’ reluctance to stay, preferring to find other locations with higher monetary benefits. This meant that RR practice in Maluku provided no promising rewards, contrasting with the recommendation from others internationally that meaningful reward for rural work is needed [67].

### 4.4. Personal and Professional Support

Although the government arrangements included substantial funding aimed at enhancing overall development in RR Indonesia, corrupt practices in critical programmes created unconducive personal and professional conditions. These practices are common in Indonesian RR districts and other LMICs [68,69,70] and underpins reasons why physicians are reluctant to remain in RR practice [41]. Furthermore, the district governments believed they already provided significant incentives; hence, the physicians were obliged to serve, whatever the living and working conditions. The professional support and development needed by all RR physicians were not considered a priority. This case study confirms the results of a qualitative study on RR physicians in Maluku [41] that concluded that corrupt governance impeded the provision of personal and professional support for RR physicians [41].

We emphasised the imperative for political actions of the district governments to enforce the implementation of multilevel programmes to improve the poor RR working and living environments. Chile [71] and Thailand [61] provide good examples of what can be achieved with government enforcement and a holistic approach. They each utilised educational, incentive-based, and regulation enforcement approaches, while the living and working environments were improved. This enhanced the recruitment and retention of the RR medical workforce in both countries [61,71]. Otherwise, the reluctance of medical graduates to remain in RR areas may continue, and the cycle of an inadequate RR workforce may remain. Medical schools could support controls to ensure the government provides standard conditions for rural training and working and potentially help oversee their implementation.

In Australia, in addition to well-established living and professional conditions in RR areas, there is professional networking and recognition of RR physicians. The Australian College of Rural and Remote Medicine equips physicians to work in RR areas by providing ‘Rural Generalist’ training that is relevant to a community’s health needs and the conditions of RR areas [72]. It maintains and enhances RR physicians’ skills, competence, and ability to sustain RR practice [72]. This success indicates the benefits expected from establishing an Indonesian collegium of RR physicians to support and guide the postgraduate education and professional development needs of Indonesian RR physicians.

### 4.5. Implications for Practice

Beyond the exclusive and seemingly sporadic, less supported interventions such as those shown by PMFU in Maluku, there is a need for an approach that comprehensively integrates all the WHO-recommended interventions combined when possible. Australia and Canada have experienced improved RR medical workforce with comprehensive interventions nationwide through ‘Rural Pipeline’ and ‘Rural Pathways’ [64,73,74,75,76,77,78,79,80]. In Australia, the government provides Rural Health Multidisciplinary Training funding to medical schools nationwide and mandates them to apply a 25% quota for rural background students, and 25% of clinical placements provided are located in rural communities [2,62,64]. Along the way, regulatory intervention with a return of services and bonded placements for medical graduates ensure that RR posts are never vacant. Further, the personal and professional needs of RR physicians are supported. Continuing medical education and professional development, such as ‘Rural Generalist’, are prioritised without the need for RR physicians to leave their posts; thus, continuity of practice is well-maintained [75,76,77]. These comprehensive and sustainable strategies are worth implementing in the Indonesian context and, therefore, need consideration at the national level. The PUFM, in partnership with some district governments, has initiated educational interventions that have been supported by district government regulation for improved recruitment. This initiative needs to be followed up and supported at various levels to create a model of rural pipelines and pathways within the Indonesian context.

### 4.6. Study Strengths and Limitations

The implications from this study are generalisable to other countries, regions, and districts as common features of rural-focused medical education are applicable across different contexts. Additionally, Maluku represents Indonesia’s and other archipelagic countries’ rural and remote conditions regarding personal and professional support. However, caution is required concerning place-based needs assessment, capability, availability, and willpower against corrupt governance. Further exploration of other RR districts in Indonesia can help elucidate the importance of context in implementing the best-evidenced strategies and interventions.

Another limitation of this study is the potential bias of the authors during the interpretation and data analysis. The authors’ experiences in medical education and the RR medical workforce could influence their interpretation of data. However, the authors maintained reflexivity and adhered to the pre-determined analysis framework.

## 5. Conclusions

Indonesia has implemented most of the WHO recommendations for improved medical workforce recruitment and retention in RR areas. However, there is no documentation of educational recommendations at national or government levels applied to medical schools nationwide. Regional medical schools have shown their commitment to social accountability in fulfilling the country’s needs for a medical workforce, yet nationwide government intervention in educational aspects is imperative to magnify the impact of medical school initiatives throughout the nation. Relevant programmes have been implemented nationally, regionally, and locally at the district level. However, it is necessary to re-evaluate and re-enforce these programmes regarding their significance, comprehensiveness, integration, and effectiveness in creating a sustainable RR medical workforce. Although place-based contexts are required to implement recommended interventions, best-evidenced comprehensive interventions should inform all efforts.

## Figures and Tables

**Table 1 ijerph-20-01435-t001:** Summarised key achievements and areas that needed improvements.

WHO-Recommended Intervention	Key Achievements	Areas Needing Improvement	
A. Education			
	Improved number of physicians in RR districts from local medical schools	Retention related to these strategies needs evaluation	
	Continuing education and professional development of RR physicians supported by medical school	No nationwide implementation of WHO educational recommendations	
		No medical college for RR physician postgraduate training and recognition	
B. Regulatory interventions			
	Partially improved recruitment of RR physicians	Enhanced scopes of practice for RR physicians through regulation and funding	
	Supported RR physicians’ professional development with scholarships and education subsidies	Regulation enforcement regarding non-adherence to return of service	
C. Financial Incentives			
	Attracted more physicians to RR areas	Different amounts of incentive received across different employment schemes despite the same workload and area remoteness	
D. Personal and professional support			
		Living conditions and basic needs (clean water supply, electricity, telecommunications, etc.)	
		Complicated bureaucracy to improve working conditions	
		Minimum career development and professional support	
Overall			
		Integration of all recommendations and comprehensiveness of strategies	
		Enforcement of regulations	

## Data Availability

Not applicable.

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
