# Peer review of "Attracting, Recruiting, and Retaining Medical Workforce: A Case Study in a Remote Province of Indonesia"

_ijerph, 2023, doi:10.3390/ijerph20021435_

Round 1
Reviewer 1 Report
The article was well-written, provides clear description about the policies implemented at national and sub-national level, focusing on a remote province in Indonesia. The article provides additional information to add to the body of knowledge about recruiting and retaining health workers in Indonesia.
Some comments are written in the pdf file. More clear methods, as to how many respondents from each category were included can give a better perspective to the reader, as many of the quotations seemed to come from the provincial health office rather than the DHO ones.

Author Response
|
Comments and suggestions for authors |
Authors responses |
|
The article was well-written, provides clear description about the policies implemented at national and sub-national level, focusing on a remote province in Indonesia. The article provides additional information to add to the body of knowledge about recruiting and retaining health workers in Indonesia.
|
The authors would like to thank the reviewer for providing supportive and constructive comments and valuable reviews. We have responded to the comments and suggestions annotated in the PDF files. -->attached
|
|
Some comments are written in the pdf file. More clear methods as to how many respondents from each category were included can give a better perspective to the reader, as many of the quotations seemed to come from the provincial health office rather than the DHO ones.
|
The methods have been edited to give a better perspective of the number of respondents from each group. Previously we mentioned this in the results section (Pg 3, lines 136-139). We have made changes in the text text-based upon all of the comments made by the reviewer in the annotated pdf manuscript The quotations also come from the DHO, as can be seen in the paragraphs explaining the quotations.
|

Reviewer 2 Report
I would like to congratulate the authors for undertaking this research and using a case study approach to get a comprehensive understanding of factors impacting medical recruitment in this area of Indonesia. I think there areas that could be improved in terms of some overstating success of some programs elsewhere and also in improving clarity of presentation.
Abbreviations were too frequently used particularly for terms that were not frequently used. Trying to remember what all the abbreviations meant or having to go back and find what the abbreviations stood for reduced readability.
In the introduction, the sentence starting on line 49 implies that it is only the medical workforce that influences health indicators and discounts things such as public health strategies and the other professions within a multidisciplinary team.
The purpose of the study could be better expressed - was is meant by 'comprehend strategies'. Perhaps understand strategies is more appropriate.
In the Materials and Methods section use past tense. It was hard to follow if the documents were sourced from the university where the medical school was located. Refer to the actual university.
Given that there were only 15 people interviewed, more information could be provided regarding who they were ie. how many medical school executive, etc.
The analysis framework needs better explaining in the data analysis section. Was a discourse analysis used to analyse the data from the documents and the interview data analysis was not clear. Narrative reporting of the interviews is not a data analysis method. This is a major weakness of the paper.
Table 1 should be included in the findings section. Table 1A is not needed as does not provide any further information than what is in the findings or discussion.
There are a number of typos and missing words - line 229 (of), 240 (level not line), 266 and 272 (brackets needed for references), line 481, line 592
Line 297 - what is meant by n years - state the actual number
Line 418 - suggest changing the wording to more appropriate expression than 'fiddling with'
In the discussion, there is an overstating of the success of Australian programs and the success of Rural Clinical Schools in terms of increasing the rural medical workforce. They have limited success in terms of rural training locations, exposure and rural-focused curriculum with only limited number of medical courses in Australia being located entirely in regional areas. On page 12, it is also misleading to claim that fulfilling 25% rural background student quota and 25% of clinical placements is a success.
There are other limitations such associated with the study design which are not mentioned.
Author Response
|
Comments and suggestions for authors |
Authors responses |
|
I would like to congratulate the authors for undertaking this research and using a case study approach to get a comprehensive understanding of factors impacting medical recruitment in this area of Indonesia. I think there areas that could be improved in terms of some overstating success of some programs elsewhere and also in improving clarity of presentation.
|
The authors would like to thank the reviewer for providing supportive and constructive comments and valuable reviews. We have edited the manuscript accordingly. (all the lines mentioned refer to the clean PDF version of the revision) |
|
Abbreviations were too frequently used particularly for terms that were not frequently used. Trying to remember what all the abbreviations meant or having to go back and find what the abbreviations stood for reduced readability.
|
Thank you for raising this concern. We have reduced the infrequently used abbreviations.
|
|
In the introduction, the sentence starting on line 49 implies that it is only the medical workforce that influences health indicators and discounts things such as public health strategies and the other professions within a multidisciplinary team.
|
We have edited the wording, as can be seen on lines 49-53 (clean version PDF file) Although Indonesia barely achieves the national standards for the underlying health indicators[5], there are limited published studies describing the approaches implemented by the Indonesian government regarding recruitment and retention of the medical and healthcare workforce. Health workforce strategies operate concurrently with other public health strategies nationally or regionally, which aim to improve population health. |
|
The purpose of the study could be better expressed - was is meant by 'comprehend strategies'. Perhaps understand strategies is more appropriate.
|
We have edited the wording Page 2 line 67-70 The purpose of this study was to document and understand strategies recommended and implemented for building a sustainable medical workforce, including the constraints on implementation of rural recruitment, development, and retention strategies in one remote province of Indonesia.
|
|
In the Materials and Methods section use past tense. It was hard to follow if the documents were sourced from the university where the medical school was located. Refer to the actual university.
|
We have edited the section accordingly (lines 95-97) Documents from the regional medical school, Pattimura University Faculty of Medicine (PUFM) were accessed through the school executives.
|
|
Given that there were only 15 people interviewed, more information could be provided regarding who they were ie. how many medical school executive, etc.
|
We have mentioned this in the results section in the previous version. We have added it to the methods section. (lines 107-111). Using an interview guide, we conducted interviews with 15 respondents including provincial health officials related to human resources (n=3), head of District Health Offices (n=7), Dean and former Dean of PUFM (n=2), head of regional Indonesian Medical Association (n=1), Puskesmas accreditation surveyors (n=2).
|
|
The analysis framework needs better explaining in the data analysis section. Was a discourse analysis used to analyse the data from the documents and the interview data analysis was not clear. Narrative reporting of the interviews is not a data analysis method. This is a major weakness of the paper.
|
Thank you for highlighting this. We have added the analysis method (lines 117-118 of the revised manuscript) Thematic analysis[17] was used for data synthesised from document analysis and the KIIs. The themes were predefined in the analysis framework.
|
|
Table 1 should be included in the findings section. Table 1A is not needed as does not provide any further information than what is in the findings or discussion.
|
We have moved Table 1 to the Results section and removed Table A1
|
|
There are a number of typos and missing words - line 229 (of), 240 (level not line), 266 and 272 (brackets needed for references), line 481, line 592
|
We have edited them accordingly
|
|
Line 297 - what is meant by n years - state the actual number
|
We cannot state the actual number, as it depends on the length of study of each specialist-training program. We have edited the wording for a better understanding of the n years (lines 316-318) They are required to serve in hospitals in remote areas that need specialists with a time bond minimum of n years and maximum of 2n years (n=length of study, it varies depending on the specialist training program).[20,31,39,40]
|
|
Line 418 - suggest changing the wording to more appropriate expression than 'fiddling with'
|
We have edited the wording (changed to modifying) Line 445 Another reason stated is due to the use of information system and report via a web-based application, there is no chance of scamming and modifying a project value.
|
|
In the discussion, there is an overstating of the success of Australian programs and the success of Rural Clinical Schools in terms of increasing the rural medical workforce. They have limited success in terms of rural training locations, exposure and rural-focused curriculum with only limited number of medical courses in Australia being located entirely in regional areas |
We have edited the wording regarding the limitation of the regional medical school. lines 520-530 Rural Clinical Schools across Australia were implemented as a national educational intervention to address the geographical inequalities of medical workforce.[2,63] At the time, in terms of rural training locations, exposure and rurally-focused curriculum, only a limited number of medical courses in Australia were located entirely in regional areas.[64] The advent of the Rural Clinical School programs means the medical schools are not acting sporadically and exclusively but rather work systematically under the considerations imposed by the national government. The predefined quota of rural background medical students in the admission program ensures the representativeness of rural communities and supports the ‘rural pipeline' program[2,65]; in the end, this approach will benefit the community by having medical graduates that are more likely to return and be retained in the community.
|
|
On page 12, it is also misleading to claim that fulfilling 25% rural background student quota and 25% of clinical placements is a success. |
Concerning the rural background student quota and clinical placements, we have edited the wording claiming success to experience an improved RR workforce. Lines 601-606 Australia and Canada have experienced improved RR medical workforce with comprehensive interventions nationwide through 'Rural Pipeline' and 'Rural Pathways'.[65,74-81] In Australia, the government provides Rural Health Multidisciplinary Training funding to medical schools nationwide and mandates them to apply a 25% quota for rural background students, and 25% of clinical placements provided are located in the rural communities.[2,63,65]
|
|
There are other limitations such associated with the study design which are not mentioned.
|
We have added the study limitation Lines 628-631
Another limitation of this study is the potential bias of the authors during the interpretation and data analysis. The authors’ experiences in medical education and the RR medical workforce could influence their interpretation of data. However, the authors have maintained reflexivity and adhered to the pre-determined analysis framework.
|